# Antagonistic Functions of Connexin 43 during the Development of Primary or Secondary Bone Tumors

**DOI:** 10.3390/biom10091240

**Published:** 2020-08-26

**Authors:** Julie Talbot, Maryne Dupuy, Sarah Morice, Françoise Rédini, Franck Verrecchia

**Affiliations:** 1INSERM/CNRS UMR3347/U1021, Curie Institute, 91405 Orsay, France; julie.talbot@curie.fr; 2INSERM UMR 1238, Nantes University, 44035 Nantes, France; maryne.dupuy@etu.univ-nantes.fr (M.D.); sarah.morice@univ-nantes.fr (S.M.); Francoise.redini@univ-nantes.fr (F.R.)

**Keywords:** gap junction, connexin 43, bone tumors, primary tumor growth, metastatic process

## Abstract

Despite research and clinical advances during recent decades, bone cancers remain a leading cause of death worldwide. There is a low survival rate for patients with primary bone tumors such as osteosarcoma and Ewing’s sarcoma or secondary bone tumors such as bone metastases from prostate carcinoma. Gap junctions are specialized plasma membrane structures consisting of transmembrane channels that directly link the cytoplasm of adjacent cells, thereby enabling the direct exchange of small signaling molecules between cells. Discoveries of human genetic disorders due to genetic mutations in gap junction proteins (connexins) and experimental data using connexin knockout mice have provided significant evidence that gap-junctional intercellular communication (Gj) is crucial for tissue function. Thus, the dysfunction of Gj may be responsible for the development of some diseases. Gj is thus a main mechanism for tumor cells to communicate with other tumor cells and their surrounding microenvironment to survive and proliferate. If it is well accepted that a low level of connexin expression favors cancer cell proliferation and therefore primary tumor development, more evidence is suggesting that a high level of connexin expression stimulates various cellular process such as intravasation, extravasation, or migration of metastatic cells. If so, connexin expression would facilitate secondary tumor dissemination. This paper discusses evidence that suggests that connexin 43 plays an antagonistic role in the development of primary bone tumors as a tumor suppressor and secondary bone tumors as a tumor promoter.

## 1. Gap Junction Channels

According to Schleiden’s theory, cells are autonomous units circumscribed by a diffusion barrier that prevents any exchange with the surrounding cells [1]. However, it appears that exchanges of molecules take place through this diffusion barrier. These observations led Höber and colleagues to hypothesize the existence of hydrophilic channels in cellular membranes [2]. In 1952, Weidmann and colleagues demonstrated that despite their multicellular structure, Purkinje cardiac fibers present a continuous electrical conduction, which suggests the existence of a cytoplasmic continuity between the cells that constitute those fibers [3]. A few years later, Kanno and Loewenstein proved that fluorescein (376 Da) can freely pass from one cell to another [4]. Two years later (1966), Loewenstein proposed that junctional communication occurs through channels that cross the membranes of the two cells in contact. Each of these channels is composed by a pair of permeable membrane units that form an intercellular channel [5]. In 1967, these structures were identified following the observation of ultra-thin sections of mouse myocardium; there was an extracellular space of 2 to 4 nm between two cells, hence the term “gap junction” [6]. Finally, in 1975, the term “communicating junctions” was used to refer to the function of these structures [7].

### 1.1. Structure

Since the late 1970s [8], gap junctions have been defined as membrane structures that allow for the direct transfer of ions, metabolites and anticancer drugs [9,10,11,12,13], from cytoplasm to the cytoplasm of adjacent cells. Studies using electron microscopy or optical diffraction approaches have proven that each intercellular channel consists of two hemichannels (called connexons), both composed of six gap junction proteins (connexins), that align head-to-head on opposing cell surfaces to form an intercellular channel [14,15] (Figure 1A). Connexons are composed of six connexins that form a central pore of ~2 nm in diameter, allowing the diffusion of small molecules with a molecular mass of less than 1200 Da, such as second messengers (e.g., calcium, inositol triphosphate, adenosine monophosphate, glutathione, glucose, glutamate) [16]. More recently, studies have hypothesized that siRNA and miRNA can also pass through Gj despite having molecular masses of ~8 kDa [17,18,19,20].

Homotypic (association of two identical connexons) or heterotypic (association of two different connexons) intercellular channels can form in the plasma membrane [21] (Figure 1B). Connexons are also able to have a function per se and act as hemichannels [22]. This activity may be extended to pannexins, which are connexin-related proteins that are expressed in many human organs and can form large-pore single channels that allow the release of ATP and other metabolites crucial for cellular communication [23,24,25].

### 1.2. Gap Junction Proteins: Connexins

Twenty-one genes that code for different connexins have been identified in the human genome, and 20 have been identified in the mouse genome [26,27]. Currently, connexin proteins and their genes are classified according to two systems of nomenclature. The first nomenclature is based on the molecular weight of connexins in kDa. For example, connexin43 is the connexin with a molecular weight of approximately 43 kDa. The second nomenclature divides connexins into subgroups (α, β, γ, δ and ε) with respect to their extent of sequence identity and length of the cytoplasmic domains [28]. In this system, connexin isoforms are labeled “Gj” and are serially numbered based on the order of their discovery [28,29].

In mammals, all connexins have a similar topology that is characterized by the presence of four hydrophobic transmembrane domains (M1 to M4) that are connected by an intracellular loop (IL) and two extracellular loops (EL-1 and EL-2) (Figure 1C). The NH2 and COOH-terminal domains are in the cytoplasm [30,31]. Biophysical and structural studies revealed that the N-terminal half of connexin protein contains the principal components of the pore and the gating elements [32,33]. The transmembrane regions M1, M2, and M4 contain hydrophobic residues, whereas the M3 domain has polar residues that enable the formation of the aqueous pore of the channel. The EL-1 and EL-2 extracellular loops contain cysteine residues that form intra-molecular disulfide bonds, which play a crucial role in the formation of the channel [34,35]. Connexins, however, differ in their IL and carboxyl terminus. The sequences and the lengths of these domains are highly variable among different connexins. Posttranslational modifications, such as S-nitrosylation, ubiquitination, sumoylation, and phosphorylation, directly regulate GJs [36,37,38,39,40,41,42]. In particular, several phosphorylation domains that are activated by kinases such as protein kinase A or protein kinase C are located on the COOH-terminal end of many connexins. These processes of connexins phosphorylation are important for channel opening and the formation and removal of gap junction channels. For example, the phosphorylation of connexin43 at specific sites can regulate its assembly, internalization and degradation [43,44].

## 2. Brief Overview of the Role of Connexin43 in Solid Cancer Development

Information exchange between cells and their environment is crucial to allow for cells to coordinate their activity. Since Gj is an essential mechanism in this exchange process, it is not unsurprising to observe that a correlation has been observed between a deregulation of Gj and the development of various pathologies such as cancers. In the late 1960s, Loewenstein and Kanno were the first to describe a decrease or even complete loss of Gj in tumor cells, specifically in liver cancer cells [5,45]. Since then, numerous studies have described a loss of Gj in many cancer cells, thereby highlighting the crucial role of Gj in cancer development. Historically, connexins were defined as tumor suppressors. However, since the last decade, the scientific community’s understanding of Gj and connexins in cancer has grown considerably, and it is now accepted that Gj and connexins have multiple functions in cancer development, from primary tumor growth to metastatic progression [12,46,47]. There are many connexins (21 in humans), and they have differential roles in tumor initiation, progression or metastasis [12,46,47,48,49,50,51]; this review focuses on the role of connexin43 in cancer development.

### 2.1. Connexin43 and Primary Tumor Growth

The role of Gj in primary tumor growth is mainly associated with its ability to inhibit cell cycle progression. Indeed, numerous studies demonstrated the crucial role of Gj in the regulation of cell proliferation. Since the late 1980s and early 1990s, it has been recognized that the ability of cells to communicate via Gj is inversely correlated with their proliferation rate [52,53,54].

The role of Gj, mediated by connexin43, in primary tumor growth is strongly supported by studies that have demonstrated a decrease and even a complete loss of connexin43 expression in primary tumors such as breast cancer or melanoma [55,56,57]. The lack of Gj mediated by connexin43 may also be due to cytoplasmic localization of connexin43, which has been observed in gastric cancer [58,59], pancreatic cancer [60,61], and breast cancer [62,63,64]. In vivo, an increase in the number and size of lung nodules has been observed in response to the carcinogen urethane in mice, where connexin43 expression was reduced [65]. Although it is well established that connexin43 can regulate cell proliferation and thus the primary growth tumor, the mechanisms by which connexin43 acts are still discussed. Two main hypotheses can be identified: an effect via mechanisms (1) that are dependent of Gj or (2) independent of Gj.

The first hypothesis is based on a wide range of data. Connexin over-expression in many Gj-deficient cancer cells leads to restored Gj and therefore a reduction or complete blockage of cell proliferation [12,46,47]. For example, connexin43 over-expression in B16-BL6 mouse melanoma enhances Gj, inhibits cell proliferation and prevents anchorage-independent growth, whereas cell migration is unaffected [66]. Moreover, connexin43 over-expression in FMS (Macrophage Colony-Stimulating Factor 1 Receptor) human melanoma cells increases basal and tumor necrosis factor-α-induced apoptosis and decreases melanoma tumor growth [67]. The hypothesis of the main role of Gj in the control of tumor growth by connexin43 is reinforced by how using chemical inhibitors of GJ lead to an increase in cell proliferation in most cellular models [68]. Interestingly, it has been proven for several decades that tumor promoters such as tetradecanoyl 12-13 phorbol acetate (TPA) inhibit Gj in most cells and thus enhance cell proliferation and primary tumor growth [69,70]. One of the mechanisms that explains the role of Gj in controlling cell proliferation is the ability of Gj to promote the diffusion of growth regulators such as calcium or cAMP between cells [68,71].

More recently, it has been proven that Gj can also promote the exchange of miRNA between cells, allowing, for example, the enhancement of the antiproliferative properties of miRNA-124-3p in glioblastoma cancer cells [72]. Interestingly, several protein kinases implicated in tumor development such as PKC (Protein Kinase C) can phosphorylate connexin43, decreasing channel opening and increasing cell proliferation [73,74,75,76]. Src activity, which is implicated in the progression of many types of cancer, can block Gj by directly phosphorylating connexin43 at tyrosine residues [77,78,79]. Furthermore, Src can recruit MAPK (mitogen-activated protein kinase) to phosphorylate connexin43 on serine residues to disrupt Gj [79,80]. Notably, over-expression of connexin43 in glioma cells decreases Src activity [71], which suggests a reciprocal loop of activation and inactivation between Src and connexin43.

Although all effects of connexin43 on cell proliferation were initially attributed to Gj, some of the connexin43 functions occur are unrelated to channel function [81]. Indeed, C-terminal tail of connexin43 interacts with protein components of mitogenic signaling cascade and the cell cycle control system, enabling connexin43 to control cell proliferation in manners that are independent of the ability of connexin to form Gj channels [82,83]. Schematically, under normal conditions, connexin43 binds and therefore sequesters some oncogenic factors involved in the regulation of cell proliferation. In pathological conditions such as cancers, the decrease of connexin43 expression leads to the release of these pro-oncogenic factors, which in turn stimulate cell proliferation. For example, the binding of the catenin with the C-terminal tail of connexin43 reduces the amount of free catenin available for Wnt signaling to regulate cyclinD1 expression and therefore cell proliferation [84,85,86]. CCN3 is also able to interact with the C-terminal domain of connexin43 [87,88]. It is thus supposed that the down-expression of connexin43 in cancer cells favors the release of CCN3, its nuclear translocation and therefore cell proliferation [68]. For Src signaling, it has been demonstrated that the C-terminal domain of connexin43 interacts with PTEN (Phosphatase and Tensin homolog) and Csk to inhibit the pro-proliferative properties of Src in glioma cells and astrocytes [89]. Finally, many of connexin 43’s effects on cell proliferation may be attributed to hemichannels, even though the link with cell proliferation has not been fully demonstrated [68].

### 2.2. Connexin43 and the Metastatic Process

Tumorigenesis is a long-term and multi-stage process that most often leads to the development of metastases. The metastatic progression is finely regulated by various crucial cellular processes, including the epithelial-to-mesenchymal transition, extravasation, or intravasation and the migration process that allows the formation of distant secondary tumor from the primary tumor.

Despite initial studies that have demonstrated a decrease in metastatic potential due to connexin43, an increasing amount of works demonstrate that connexin43 expression may potentiate the development of metastases. The specific case of the role of connexin43 in the development of bone metastases from prostate cancer are discussed in chapter 3.

#### 2.2.1. Connexin43 and Epithelial-to-Mesenchymal Transition (EMT)

It is generally accepted that EMT decreases epithelial properties, increases mesenchymal properties, promotes the invasiveness of cancer cells and contributes to the development of circulating tumor cells [90]. This cellular process involves molecular and cellular modifications, including a loss of cell-to-cell interactions associated with a lack of or down-regulation of crucial components in the intercellular junction such as E-cadherin, claudins, occludins, or desmosomes. In association, there have been observations of the up-regulation of mesenchymal marker expression, such as N-cadherin, fibronectin, and vimentin. These two events are closely regulated by different transcription factors such as Snail-1, Snail-2 (Slug), ZEB-1 and ZEB-2, or Twist [90]. EMT is also triggered by a wide variety of growth factors such as TGF-βs, Wnts, or Hedgehog proteins [91,92].

Regarding the role of connexins in EMT, it is frequently observed that EMT is accompanied not only by a down-expression of E-cadherin but also by connexins [68,93]. It has been demonstrated that connexin43 reverses the resistance of A549 lung adenocarcinoma cells to cisplatin by inhibiting EMT [94]. Down-expression of connexin43 using shRNA reduces the expression of epithelial markers such as E-cadherin and enhances the expression of mesenchymal markers such as N-cadherin and ZO-1 in MDA-MB-231 tumor cells [63]. Although the molecular mechanisms that allow the connexins to regulate EMT have not yet fully described, there are some data that links connexin and EMT-inducers such as certain transcriptional factors or different signaling pathways. Functional links between connexin43 and Snail-1 have thus been established in prostate cancer cells [95]. In addition, Hills and colleagues provide compelling evidence that TGF-β1-induced EMT leads to a loss of E-cadherin and connexin43-mediated cell communication in HEK (Human Embryonic Kidney) cells [96]. Moreover, it has been established that connexin32 stimulates EMT in hepatocellular carcinoma cells by regulating Snail expression via the Wnt/β-catenin signaling pathway [97].

#### 2.2.2. Connexin43 and Intravasation/Extravasation Process

During the metastatic process, cancer cells disseminate by entering the bloodstream, a process that is called intravasation. Then, tumor cells extravasate at metastatic sites by attaching to endothelial cells that line blood vessels and by crossing the vessel walls. The intravasation and extravasation of cancer cells require a short period, during which tumor cells are in close contact with endothelial cells before the process of transendothelial migration (TEM) [98].

It has been demonstrated that lymphocyte gap junctions are involved in intercellular communication during migration of lymphocytes across the endothelium of central or peripheral tissues [99]. Regarding cancers, it seems that connexin43 facilitates metastatic “homing” by increasing adhesion of cancer cells to endothelial cells. Several studies concerning this process have been described with respect to metastatic breast cancer, such as HBL100 mammary epithelial tumor cells that express connexin43 have been observed to form-functional heterocellular gap junctions with HMVEC (Human Microvascular Endothelial Cell) monolayers, which allows diapedesis [100]. Elzarrad and colleagues demonstrated that the adhesion of breast cancer cells to the pulmonary endothelium increases when cancer cells overexpress connexin43 and markedly decrease when cells expressing dominant-negative connexin-43 [101]. Metastatic 4T-1 breast cancer cells express connexin43 and form-functional Gj, with the endothelium increasing the extravasation and brain colonization [102].

#### 2.2.3. Connexin43 and Migration Process

Tumor cell migration is an essential process for the development of metastases. Schematically, tumor cells extend filopodia or lamellipodia to the migration front, adhere to the extracellular matrix (ECM) proteins and form focal adhesions to allow tumor cells movement. The orchestration and synchronization of each step of collective or single cell migration require interactions between migrated cells and between migrated cells and their cell environment. Several proteins related to cell-cell communication, including connexins, regulate this migration process [103] either through Gj, allowing direct cell-cell communication, or through hemichannels that provide a pathway for the release and uptake of small molecules to or from extracellular compartments [104].

Connexin43 participates in the migration of many cells such as astrocytes [105,106,107]. For example, connexin43 is required for astrocyte cell migration under pro-inflammatory conditions [108]. Connexins have also been proven to play a crucial role in cell migration in other cell types such as neurons [109,110], keratinocytes [111], endothelial cells [112] and bone marrow stromal cells [113].

The first indication that connexin43 stimulates migration of tumor cells was the observed over-expression of connexin43 in cervical carcinoma HeLa cells [114]. Another work has used breast MCF-10A cells to demonstrate the role of connexin43 in tumor cells migration. In these cells, connexin43 controls migration because the knockdown of connexin43 leads to erratic, slow, and reverse migration. One hypothesis to explain this process is that the down-expression of connexin43 could increase the ability of MCF-10A cells to form protrusions, which results in cells with a more polygonal shape decreasing their ability to migrate [115]. Even if the role of connexin43 in cancer cell migration still remains controversial, an increasing amount of data indicates that connexins favors the migration of tumors cells such as rat hepatocellular carcinoma cells [116], glioma cells [117,118,119,120,121], liver cancer cells [116] or WEHI 231 B lymphoma cells [122].

#### 2.2.4. Connexin43 and Long-Distance Communication via Extracellular Vesicles (EVs)

EVs are well known as crucial vehicles able to drive the exchange of information across distant cells in many biological processes such as cancer progression [123,124,125,126]. During recent years proteomic studies identified connexins including connexin43, connexin45 and connexin32 in EVs [127,128,129]. It has been proposed that connexin43 channels at the EV surface associate with hemichannels at the plasma membranes of the target cells and forms a Gj-like structure able to transfer small molecules [128,129,130] playing thus a crucial role in cancer development [131]. In this context, bioinformatics analysis suggests that RNA- and DNA-binding described in connexin43 sequences might be important to consider for studies on transfer of genetic information EVs [132].

## 3. The Role of Connexin43 in Primary and Secondary Bone Tumors

Bone tumors may be classified as "primary” tumors, which originate in bone, and "secondary” tumors that arise as a result of soft tissue tumors metastasizing to the bones.

### 3.1. Connexin43 and Bone Remodeling

Bone remodeling plays a crucial role in bone tumor development. A vicious cycle between tumor cells and bone cells has been described during bone tumor development. In brief, cancer cells produce soluble factors such as osteolytic cytokines that activate osteoclastogenesis, leading to bone degradation. Following bone degradation, pro-oncogenic factors that are trapped into the bone matrix are released in the bone microenvironment and thus stimulate tumor growth [133,134].

#### 3.1.1. Brief Overview about Bone Remodeling

Bone tissue is the site of an ongoing remodeling that mainly involves two cell lineages: the mesenchymal osteoblastic and the hematopoietic osteoclastic lineages. Schematically, the osteoblasts and the osteoclasts drive the bone formation and resorption, respectively.

Osteoblastic progenitors are mesenchymal stem cells (MSCs) [135] that are mainly present in the bone marrow and more specifically multipotent skeletal stem cells (MSSCs), which are a subset of MSCs that have been recently identified [136]. MCS cells undergo a series of proliferation and differentiation steps, leading to the formation of mature osteoblasts under the control of transcription factors such as Runx2 or Osterix [137] (Figure 2A). Runx2 is a transcriptional factor that enhances osteoblastic genes expression by binding a consensus site present along the proximal promoters of genes, including an α1 chain of type I collagen, bone sialoprotein, osteocalcin and osteopontin [138]. Interestingly, the deletion of Runx2 in mice results in a complete lack of bone formation [139]. Osterix is a transcription factor that can enhance the expression of oteoblastic genes, such as those of the α1 chain of type I collagen and osteopontin, by binding the GC-rich regions present on the proximal promoter of these genes. Deletion of Osterix in mice thus leads to major defects in bone formation [140].

Osteoclast precursors are cells of hematopoietic origin [141] that differentiate into osteoclasts in response to different transcription factors such as NF-kB and NFATc1 (Nuclear Factor of Activated T-cells cytoplasmic 1) and growth factors such as M-CSF (Macrophage Stimulating Factor) and RANK-L (Receptor Activator of NF-kB Ligand) [134] (Figure 2B). In particular, the molecular triad RANK (Receptor Activator of NF-kB), RANK-L and OPG (OsteoProteGerin) (RANK/RANK-L/OPG) plays a crucial role in the osteoclast differentiation process. Schematically, RANK-L favors osteoclast differentiation and thus bone degradation, while OPG, the decoy receptor of RANK-L, inhibits bone degradation mainly by preventing the binding of RANK-L to its receptor RANK [142].

#### 3.1.2. Connexin43 and Bone Remodeling

Development and remodeling of bone tissue requires the coordinated activity of bone cells. This coordination is controlled by many factors, including cytokines; growth factors present in the bone microenvironment; and the ability of cells to directly communicate with each other, especially by the direct cell-to-cell interactions mediated by Gj [143,144,145].

Various strategies such as histological approaches have demonstrated the presence of gap junction proteins in all cells of the bone tissue, including osteoblasts, osteoclasts and osteocytes (Figure 2C). Among the identified connexins (connexin43, connexin45 and connexin46), connexin43 is the most abundant and therefore plays a crucial role in signal transmission between different bone cells, thereby regulating the bone’s development, differentiation, modeling and remodeling [146].

The crucial role of connexin43 in skeletal development or bone remodeling was demonstrated in vivo using the connexin43-null mouse models. Deletion of connexin43 in null mice leads to the delayed intramembranous and endochondral ossification of the cranial vault in mice embryos. In addition, connexin43-deficient embryos exhibit retarded ossification of the clavicles, ribs, vertebrae, and limbs. These processes are closely associated with abnormal skeletal mineralization and osteoblast differentiation [147,148]. Mice with selective osteoblast deletion of connexin43 exhibit low peak bone and reduced bone formation and defective osteoblast function, which demonstrates the main role of connexin43 in osteogenesis [149,150].

It is now well established that Gj as mediated by connexin43 plays a crucial role in the terminal differentiation of pre-osteoblasts into mature osteoblasts and in their activity. For example, the inhibition of junctional permeability by heptanol or by the over-expression of a dominant-negative of connexin43 expression reduces the expression of osteoblastic markers such as the α1 chain of type I collagen, alkaline phosphatase, osteocalcin or bone sialoprotein [151,152]. More recently, it has been demonstrated that connexin43, whose expression is increased during the differentiation of human MSC into osteoblasts, plays a central role in osteoblastogenesis from the early stages to the process of mineralization. These observations suggest that connexin43 plays a role during the commitment of human MSC toward osteoblastic lineage [153]. Studies have examined the molecular mechanisms by which connexin43 drives the expression of osteoblast markers such as the 1 unit of type I collagen. These studies demonstrate that Gj mediated by connexin43 in osteoblast cells allows the transmission of cytoplasmic mediators. Some of these mediators can stimulate the ERK (Extracellular signal-Regulated Kinase) signaling pathway, which activates the transcription factor Sp1 that, in turn, stimulates the transcription of α1 chain of type I collagen gene [154]. More recently, the crucial role of cAMP in the ability of connexin43 to drive osteoblastogenesis has been demonstrated. Gj as mediated by connexin43 in osteoblast cells allows the diffusion of cAMP, which can modulate the expression of sclerostin, and RANK-L, two critically important factors for the regulation of bone homeostasis [9].

With respect to bone resorption, although the expression of the connexin43 has been demonstrated in osteoclasts since the 2000s [155], only a few studies have demonstrated the role of Gj in osteoclastogenesis to date. It seems that intercellular communication mediated by connexin43 plays a role in the fusion of monocyte precursors to form multinucleated osteoclasts. Blocking Gj using a chemical inhibitor or connexin-mimetic peptides affects the fusion and therefore the differentiation of osteoclast precursors into mature osteoclasts [156]. Notably, it has been recently demonstrated that another connexin, connexin37, plays a role in the regulation of osteoclast differentiation via the modulation of Gj [157].

In humans, the crucial role of connexin43 in the development of the skeleton has been demonstrated by craniofacial and limb abnormalities that have been described in patients with oculodentodigital dysplasia (ODDD), which is characterized by the presence of mutations of the gene encoding the connexin43. To date, at least 76 mutations of connexin43 have been linked to ODDD [158,159,160].

### 3.2. Connexin43 in Primary Bone Cancers

Malignant primary bone tumors include multiple myeloma, osteosarcoma, adamantinoma, chondrosarcoma, chordoma, Ewing’s sarcoma, fibrosarcoma, and undifferentiated pleomorphic sarcoma, lymphoma of bone, and malignant giant cell tumors. These bone tumors represent less than 1% of cancers and about 10% of all childhood and young adult tumors. This chapter focuses on osteosarcoma and Ewing’s sarcoma as examples of primary bone tumors.

#### 3.2.1. Generalities about Osteosarcoma and Ewing’s Sarcoma

Osteosarcoma (OS) and Ewing’s sarcoma (ES) represent the two most common malignant primary bone tumors observed in children, adolescents, and young adults (median age occurrence of 18 and 15 years, respectively, for OS and ES).

Among pediatric cancers, OS ranks eighth after lymphomas and brain tumors, with an incidence of three in one million per year. About 80% of OS develop around the knee (distal femur more often than proximal tibia) or at the metaphysis of other long bones. Osteosarcoma produces malignant osteoids (immature bone) from tumor bone cells. At diagnosis, about 20% of patients present metastases, most commonly in lungs, but also in bone or lymph nodes. Whole-genome sequencing analyses have demonstrated that OS displays high rates of genetic alterations and contain many somatic mutations and copy number alterations [92,161,162,163].

After OS, ES is the second most common malignant primary bone tumor in children, adolescents, and young adults. The annual incidence is estimated at 1–3 in one million. ES usually occurs in bone sites, with a predominance of long bones (femur, tibia, fibula and humerus), or in the pelvis (26%), chest wall, and spine. Approximately 25% of patients exhibit lung metastases at diagnosis. The translocation t(11;22)(q24;q12) between the FLI1 and EWS (EWing Sarcoma) genes is found in 85% of cases, giving rise to the transcription factor EWS-FLI1. The resultant EWS-FLI1 protein specifically recognizes FLI1’s DNA-binding domain and modulates the expression of target genes involved in cell proliferation and metastatic dissemination [92,161,162,163].

#### 3.2.2. Connexine43 in Osteosarcoma and Ewing’s Sarcoma

It is well established that a high-level expression of connexin43 represses OS cell proliferation and therefore primary tumor growth. Indeed, the regulation of connexin43 expression via various mechanisms drives OS cell proliferation in human U2OS cell lines [164,165,166] or in rat UMR106 cell lines [167], for example.

One study proved the major role of p27 in this process [164]. Indeed, the authors demonstrated that a forced expression of connexin43 inhibits the proliferation of U2OS cells through a reduced rate of G1/S transition of the cell cycle. In addition, they demonstrated that this cell cycle suppression was linked to the inhibition of phosphorylation of the protein Rb by the kinases CDK2 and CDK4. Finally, they proved that enforced connexin43 expression elevates the levels of p27 proteins through post-transcriptional regulatory mechanisms in which connexin43-driven Gj is the main instigator via cAMP diffusion.

More recently, the role of the Wnt signaling pathway has been demonstrated [165]. These authors demonstrated that the knockdown of connexin43 activated the Wnt/β-catenin signaling pathway, promoted proliferation and inhibited apoptosis of U2OS cell lines. Moreover, the authors demonstrated that the anti-tumor activity of resveratrol against U2OS cells occurs through up-regulating connexin43 and suppressing the Wnt/β-catenin signaling pathway. Similarly, connexin43 has also been implicated in the response of OS cells to other anti-tumor agents. For example, it has been proven that the Coleusin Factor (also named FSK88) regulates the expression of connexin43, restores the impaired Gj of OS tumor cells and therefore inhibits OS cell proliferation [167].

Interestingly, the intercellular communication via Gj between U2OS cells and osteoblasts seems to play a crucial role in OS progression because it has been demonstrated that this heterocellular communication drives the TGF-β-induced EMT in U2OS cells [168].

There is almost no published data on ES. In this context, Talbot and colleagues have proven for the first time the crucial role of connexin43 in the growth of the primary ES tumor. First, they identified a loss of connexin43 expression in ES cell lines. Second, they demonstrated that EWS-FLI1 regulates both connexin43 expression and Gj mediated by connexin43 in ES cells. Third, they used an osteolytic preclinical model of ES to demonstrate that the over-expression of connexin43 in ES cells dramatically reduces tumor growth, leading to a significant increase in animal survival. In vitro assays indicated that connexin43 over-expression increases p27 level with an associated marked decrease of Rb phosphorylation that is consistent with the observed blockade of the cell cycle in G0/G1 phase. Finally, they demonstrated that over-expression of connexin43 in ES cells modifies the link between tumor cells and osteoclasts. Indeed, the bone microarchitectural parameters, as assessed by micro-CT analysis, demonstrated an increased bone volume when connexin43 expression was enhanced. Histological analysis demonstrated that the over-expression of connexin43 in ES tumor cells inhibits osteoclast activity and therefore, bone resorption stimulating primary tumor growth [169].

It appears that connexin43 plays a crucial role in OS and ES tumor growth and can be considered to be a tumor suppressor in the growth of these primary bone tumors (Figure 3A).

### 3.3. Connexin43 in Secondary Bone Tumors

Carcinoma of the prostate, breasts, lungs, thyroid, and kidneys are the carcinomas that most commonly metastasize to bone. This chapter focuses on the bone metastasis of prostate cancer as example of a secondary bone tumor.

#### 3.3.1. Generalities about Bone Metastatic Cancer from Prostate Cancer

Prostate cancer is the most common non-cutaneous cancer in men, with an estimated 1,600,000 cases and 366,000 deaths annually worldwide [170]. This cancer is a heterogeneous group of tumors that corresponds to adenocarcinoma that most often originates from the acinar glands and ducts [171]. Despite the high long-term survival of localized prostate cancer, metastatic prostate cancer remains largely incurable even after intensive multimodal therapy [172] and is therefore the main cause of prostate-cancer-related mortality [173]. As other carcinomas, localized prostate cancer undergoes different steps in the metastatic process, leading to the colonization of secondary sites such as bones. Each step in this process of the disease (EMT, intravasation/extravasation, migration) leads to bone metastases formation [174]. Prostate cancer cells that spread out of the prostate gland exhibit an exquisite tropism for the bone to develop bone metastases. This process is driven by the interaction between invading tumor cells, bone-forming osteoblasts, and bone-resorbing osteoclasts, leading to the development of osteolytic, osteoblastic, or mixed bone lesions [175].

#### 3.3.2. Connexine43 in Bone Metastases from Prostate Cancer

A reduction of connexin43 expression was initially demonstrated in neoplastic prostatic tissue compared to normal tissue [176]. In parallel, in vitro data using tumor cell lines demonstrated a decrease or a lack of connexin43 expression in most prostate cancer cells, which were compared to normal prostatic cells [177,178]. It is now well accepted that connexin43 exerts a negative control on the primary tumor growth of prostate carcinoma and is therefore considered to be a tumor suppressor gene during the early stages of the development of prostate carcinoma, especially during primary tumor growth (for reviews, see [174,179]).

In contrast, during the later stages of prostate carcinoma development, particularly during the development of secondary bone tumors, namely bone metastases, connexin43 appears to favor tumor dissemination (Figure 3B).

Once the process of EMT is complete, one of the key events of metastatic development and therefore secondary bone tumor formation is the ability of tumor cells to migrate. In this context, it has been demonstrated that there is a correlation between the ability of connexin43 to form-functional intercellular channels and the ability of prostate tumor cells to migrate and invade [180]. Interestingly, Cronier and colleagues demonstrated that the localization of the connexin43 in the plasma membrane is crucial for its ability to drive the migration and invasion of prostate tumor cells [181]. Indeed, the authors observed an increase in the ability of cells to migrate and invade when connexin43 was overexpressed in the membrane of LNCap cells. In contrast, when the over-expression of connexin43 in PC-3 tumor cells was mainly limited to the cytoplasm, it failed to increase cell migration and invasion [181]. In this context, Szpak and colleagues postulated that migration of DU-145 prostate cancer cells expressing elevated levels of connexin43 may be crucial for the "leading front" formation during cancer invasion [182].

The hypothesis of the crucial role of connexin43 to drive the ability of prostate tumor cells to migrate has been reinforced in a more recent study. This study proved that connexin43 expression is associated with increased malignancy of prostate cancer cells [183]. The molecular mechanisms by which connexin43 drives cells migration are not fully understood. One hypothesis involves the role of connexin43 in cytoskeleton reorganization. In this context, various studies have demonstrated physical interactions between connexin43 and various cytoskeletal proteins implicated in cell migration [108]. Interestingly, Cronier and colleagues have demonstrated that connexin43 that over-expresses LNCaP cells exhibits a high level of phosphorylated Rac1, a member of the Rho-family GTPase, which is mainly implicated in the regulation of the focal adhesion kinase activity and therefore cell migration [174]. To invade, tumor cells not only need to migrate but also to degrade the extracellular matrix via the activity of tumor-associated proteinases such as MMPs (matrix metalloproteinases). In this context, Cronier and colleagues have demonstrated that a broad spectrum MMPs inhibitor, GM6001, inhibits connexin43-driven LNCaP cells to invade through Matrigel [181]. Interestingly, a link between connexin activity and MMP activation has been described [184,185]. In this context, in silico analysis demonstrates that MMPs can mediate cleavage of the C-terminal domain of connexin43 [186]. In this context, we can hypothesize that there is a control loop between the connexin43 which regulate the ability of tumor cells to invade via the expression or the activity of MMPs and the MMPs which participate in the cleavage and thus the activity of the connexin43.

It is now well established that connexin43 promotes the formation of bone metastasis by promoting one of the major stages in this process namely cell migration. Since bone is the most frequently site of prostate metastasis, one question remains unanswered: does connexin43 play a role in addressing tumor cells to bone? One of the answers would be the role of the osteogenic microenvironment niche which promotes cancer cell proliferation and bone metastasis progression. In this context, it was recently discovered that the bone niche serves as a calcium reservoir for cancer cells through gap junction and thus drive the metastatic process during the early-stage bone colonization [187].

## 4. Conclusions and Clinical Feature

The main observations suggest that connexin43 acts as a tumor suppressor gene in the early stages of bone tumorigenesis, as demonstrated during the growth of primary bone tumors such as OS and ES. In contrast, accumulating evidence suggests that connexin43 acts as a tumor promoter gene during the later stages of bone tumorigenesis, as indicated during the progression of secondary bone tumors such as bone metastases of prostate carcinoma. Intuitively, targeting connexin43 to limit bone tumorigenesis appeared to be a more promising therapeutic option in primary bone tumors than in secondary bone tumors.

Connexin43-mediated Gj can be blocked by several compounds that could be used as anticancer drugs [188]. Of note, different strategies have been considered to target connexin or Gj, and different compounds have thus been developed. Historically, 18-alpha-glycyrrhetinic and 18-beta-glycyrrhetinic were among the first compounds that were defined as able to block Gj. Although the action of these compounds has not been comprehensively described, it has been suggested that they bind to Gj channels, leading to a conformation alteration of the intercellular channels [189]. Long-chain alcohols such as heptanol and octanol have been reported to inhibit Gj in cardiac cells, for example [190]. This effect has been associated with deleterious effects on membrane fluidity [191,192]. Halothane and enflurane, both halogenated anesthetics, are able to inhibit Gj in many cellular systems such as crayfish axons [193] or astrocytes [194] with underlying mechanisms that are similar to those of heptanol or octanol [195].

Unfortunately, it has been regularly demonstrated that these compounds are not specific to connexins or Gj. It has, for example, been proven that most of them can inhibit other ionic conductances such as sodium, calcium or potassium channels [188]. In this context, tools such as connexin-mimetic peptides have been developed from the 1990s [196]. The most frequently used, the undecapeptide Gap27, which targets the EL2 sequence of connexins such as connexin43, connexin37 and connexin40 [197], can block Gj. Gap26, which can target EL1 sequence of connexin43, has also been developed [198]. Of note, it has been demonstrated that these connexin-mimetic peptides Gap27/26 are also able to alter connexin-hemichannel functions prior to Gj inhibition [196]. Other peptides that can target both Gj and connexin-hemichannels such as peptide5 or Gap19 have since been developed [199,200]. Another strategy to specifically inhibit Gj involves the development of connexin antibodies. Using fragments of antibodies such as Gap13 or Gap15 to target the C-terminus regions of connexin43 has demonstrated their efficiency at blocking Gj [201]. Interestingly, considering that connexin43 has also been reported to promote the bystander effect of chemotherapeutic drugs, targeting connexin43 could combined with treatment for chemotherapy.

## Figures and Tables

**Figure 1 biomolecules-10-01240-f001:**
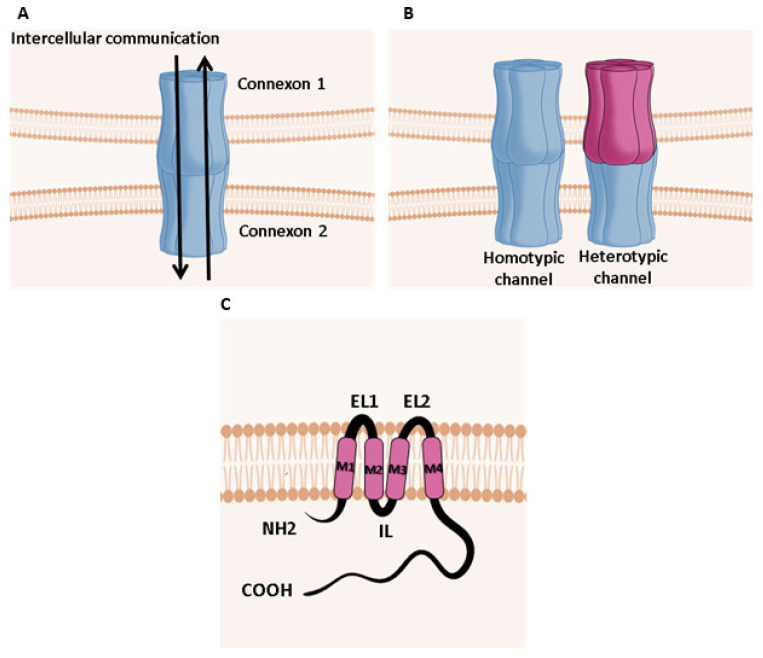
Gap junction. (**A**) Schematic representation of gap-junctional channel. (**B**) Schematic representation of homotypic and heterotypic gap-junctional channel. (**C**) Schematic representation of the connexin structure.

**Figure 2 biomolecules-10-01240-f002:**
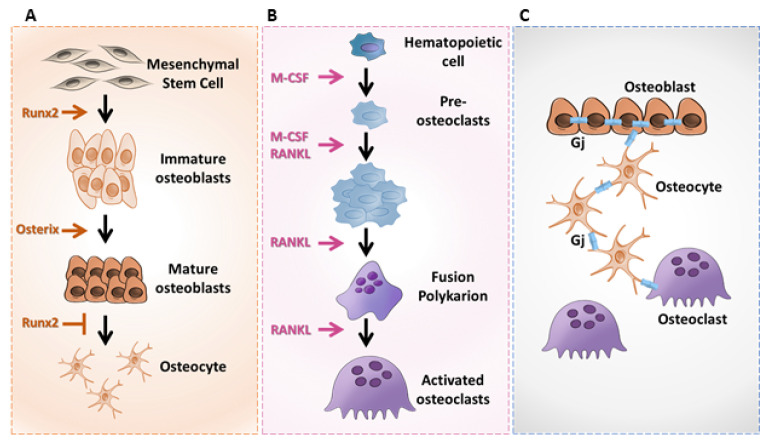
(**A**) Osteoblast differentiation. (**B**) Osteoclast differentiation. (**C**) Schematic representation of the gap junction communication (Gj) between bone cells.

**Figure 3 biomolecules-10-01240-f003:**
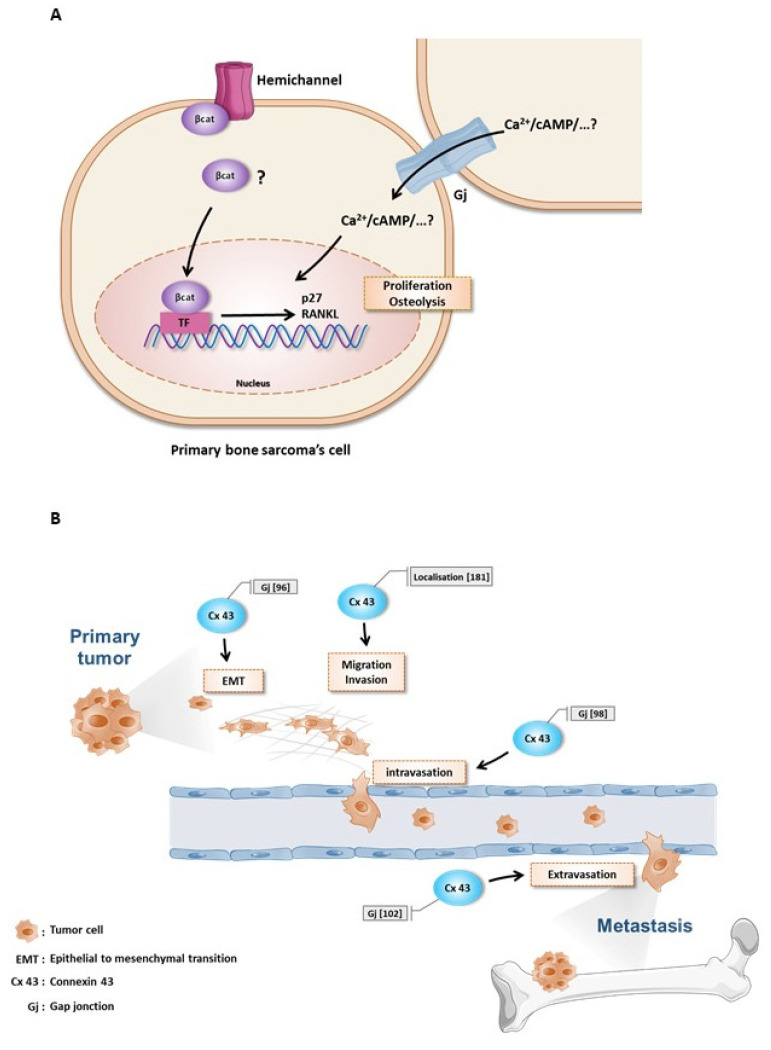
Implication of connexin 43 in (**A**) primary bone tumors and (**B**) secondary bone tumor from prostate cancer.

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
