# Peer review of "Antagonistic Functions of Connexin 43 during the Development of Primary or Secondary Bone Tumors"

_biomolecules, 2020, doi:10.3390/biom10091240_

Round 1

Reviewer 1 Report

This is a well written and organized rew about the role of Cx43 in bone cancer. However, there are some point that have to be revised:

1.- In 1.2. Gap junction proteins: connexins: There is evidence that N-terminal is located into the channel pore. Can you make a comment about it.

2.- In reference 35, Please add the following papers:
doi: 10.1186/s12860-016-0099-3
doi: 10.1039/c1mb05294b
doi: 10.3390/ijms19051296

3.- In figure 3, what about heteremeric channels ? why hemichannels are not represented ?

4.- Page 4, lines 147-163. In this paragraph it is not clear whether the authors are refereing to the C-terminal-mediated protein interactions when it is attached to the Cx43 or when is a free C-terminal (doi: 10.1016/j.bbamem.2017.05.008.)...I think you have to talk about this relatively new concept of how the Cx43 free C-terminus can have there own protein interactions and therefore own cell functions.

5.- Page 6. lines 232-242. Extracellular vesicles are very important in all these processes and also in creation of metastatic niches. Both, Cx43 and Cx46 have been observed in extracellular vesicles, I suggeste to include these papers.

6.- In figure 3, please change CNX for Cx

7.- Page 12, line 453. in "cells tumor cells not only need to" delete the first cells.

8.- Page 12, Because these is a bone Rew, I strongly suggest to add works about Cx43, MMPs and bone (alternatively I found information in osteoarthritis)

9.- In figure 3B, letters are too small and I suggest to add some detail of the role of Cx43 in each step, I mean whether it is related to hemichannels, GJCs or intracellular localization.

Author Response

Thank you for your comments about our manuscript entitled: « Antagonistic functions of connexin 43 during the development of primary or secondary bone tumors». We are very pleased with your comments that our paper is a well written and organized rew about the role of Cx43 in bone cancer. Please find herein our responses to each comments made.

  1. New sentences has been added in the section 1.2 regarding the localization of the N-terminal in the central pore.Biophysical and structural studies revealed that the N-terminal half of connexin protein contains the principal components of the pore and the gating elements (Kronengold et al., J Membr Biol, 2012; Oshima FEBS Letters, 2014)».

  1. As requested, new references have been added : Pogoda et al., BMC Molecular and Cell Biology, 2015; Schalper et al., Molecular BioSystems, 2012 and Aasen et al., International Journal of Molecular Sciences, 2018.

  1. In figure 3 : We have no data regarding heteromeric channels and primary bone tumor. Hemichannels are now represented in green.

  1. To clarify the message, a new sentence has been added « C-terminal tail of connexin43 interacts with protein components of mitogenic signaling cascade and the cell cycle control system, enabling connexin43 to control cell proliferation in manners that are independent of the ability of connexin to form Gj channels (Leithe et al., 2018) » New references have been added regarding :

-) the β-catenin : Ai et al., J Clin Invest, 2000; Sirnes et al., Int J Cancer, 2012; Moorer et al., J Cell Sci 2017.

-) CCN3 : Fu et al., J Biol Chem, 2004; Gelhaus et al., J Biol Chem 2004.

-) Src : Gonzalez-Sanchez et al., Oncotarget, 2016.

  1. Regarding Evs, a new paragraph has been added.

« 2.2.4. Connexin43 and long distance communication via extracellular vesicles (Evs)

Evs are well known as crucial vehicles able to drive the exchange of information across distant cells in many biological process such as cancer progression (Colombo et al., Annu Rev Cell Dev Biol 2014; Elzanowska et al., Mol Oncol 2020; Giordano et al., Cancers 2020Menck et al., Int J Mol Sci 2020). During the last years proteomic studies identified connexins including connexin43, connexin45 and connexin32 in EVs (Ribeiro-Rodriges et al., J Cell Sci 2017; Soares et al., Sci Rep 2015; Martins-Marques et al., 2019). It has been proposed that connexin43 channels at the Evs surface associate with hemichannels at the plasma membranes of the target cells and forms a Gj like struture able to transfer small molecules (Soares et al., Sci Rep 2015; Martins-Marques et al., 2019; Martins-Marques et al., J Extracell Vesicles 2016) playing thus a crucial role in cancer development (Zhang and You, Biochim Biophys Acta Rev Cancer 2019). »

  1. In figure 3, CNX has been changed

  1. Line 484, in « cells tumor cells not only need….», the first « cell » has been deleted

  1. Regarding MMP and connexin, new sentences have been added. « Interestingly, a link between connexin activity and MMP activation has been described (Wang et al., Molecular and Cellular Biochemistry 2014; Peng et al., European Journal of Pharmacology, 2010). In this context, in silico analysis additionally demonstrates that MMPs are able to mediate cleavage of the C-terminal domain of connexin43 (De Bock et al., Mediator of inflammation, 2015). In this context we can hypothesize that there is a control loop between the connexin43 which regulate the ability of tumor cells to invade via the expression or the activity of MMPs and the MMPs which participate in the cleavage and thus the activity of the connexin43.».

  1. Regarding figures 3B, changes have been made as requested by reviewer.

  1. New sentences have been added regarding the role of the bone niche. « Since bone is the most frequently site of prostate metastasis, one question remains unanswered: does connexin43 play a role in addressing tumor cells to bone? One of the answers would be the role of the osteogenic microenvironment niche which promotes cancer cell proliferation and bone metastasis progression. In this context, it was recently discovered that the bone niche serves as a calcium reservoir for cancer cells through gap junction and thus drive the metastatic process during the early-stage bone colonization (Wang et al., Cancer Cell 2018) ».

We hope that all the modifications made to our original manuscript address the reviewers’ issues satisfactorily and that this revised version is now acceptable for publication.

Sincerely,

Franck Verrecchia, PhD

Reviewer 2 Report

This is a solid review of Connexin43 in primary and secondary bone tumors.  The major concern is that the review is dated.  For instance, by a brief tabulation of the bibliography, of 179 references, 3 are from 2020 (one original manuscript and two reviews) and eight are from 2019 (two original manuscripts and six reviews).  Thus this reviewer could find that the last one and a half years of published research is summarized by 3 original manuscripts which is less than 2% of the literature cited.

A topical review ideally should be an index of recent studies, connecting the conclusions or identifying controversies in differing results.

The authors are encouraged to use their extensive knowledge of Connexin43 and bone tumors, as well as Connexin43 and cancer in general, to update this review so that it is a more topical summary of recent studies.

Author Response

Thank you for your comments about our manuscript entitled: « Antagonistic functions of connexin 43 during the development of primary or secondary bone tumors». We are very pleased with your comments that our paper is a solid review of connexin43 in primary and secondary bone tumors. As requested, we have now included more recent references (less than 5 years old) and removed some old references (over 5 years old).

To summarize, there are now 42% (86/201) of references published in or after 2015, and 22% (45/201) of references published in or after 2018.

References added:

1.1        Laird and Lampe, Nat Rev Drug Discov, 2018

Al Deen et al., Front Med, 2019

Umrani et al., Islets, 2017

Hong et al., Oncotarget, 2015

Peng et al., Cancer Sci, 2019

1.2       Alai et al., BMC Cell Biol 2018

Sun et al., J Biol Chem 2018

Totland et al., Cell Mol Life Sci, 2020

Pogoda et al., BMC Molecular and Cell Biology, 2016

Aasen et al., International Journal of Molecular Sciences, 2018

  1.  

Bonacquisti and Nguyen, Cancer Lett 2019

Zefferino et al., Cells, 2019

Tschrnig, Cancers 2019

Aasen et al., Oncogene, 2019

Bonacquisti and Nguyen, Cancer Lett 2019

2.1        Aasen, Cell Tissue Res 2015

Li et al., Arch Med Sci 2020

Georgikou et al., Cancer Lett 2020

Sinha et al., Trends Cancer 2020

Leithe et al., 2018

Fostok et al., J Mammary Gland Biol Neoplasia, 2019

Moorer et al., J Cell Sci 2017

2.2.4     Elzanowska et al., Mol Oncol 2020

Giordano et al., Cancers 2020

Menck et al., Int J Mol Sci 2020

Ribeiro-Rodriges et al., J Cell Sci 2017

Soares et al., Sci Rep 2015

Martins-Marques et al., 2019

Martins-Marques et al., J Extracell Vesicles 2016

Zhang and You, Biochim Biophys Acta Rev Cancer 2019

Valera-Eirin et al., BBA Mol Cell Res 2017

3.1.1     Vimalraj et al., Int. J. Biol. Macromol. 2015

            Wu et al., Tumour Biol. 2015

3.1.2     Buo et al., J. Bone Miner. Res. 2017

3.2.2     Asencio-Barria, Cancers 2019

De Bock et al., Mediator of inflammation, 2015

Wang et al., Cancer cell 2018

References removed

1.1       Boitano et al., Cell Calcium 1998

                        Evans et al., Mol Membr. Biol. 2002

Yeager et al., Curr. Opin. Struct. Biol 1996

Lim et al., Cancer Res 2011

Brink et al., Biochim. Biophys. Acta 2012

Panchin et al., Curr. Biol. 2000

Bao et al., FEBS Lett. 2004

1.2       Oshima A., FEBS Lett 2014

2          Cronier et al., Antioxid. Redox Signal 2009

Naus and Laird Nat. Rev. Cancer 2010

2.1       Goodall and Maro, J. Cell Biol 1986

                        Ai et al., J. Clin. Invest. 2000

2.2.3     Homkajorn et al., Neurosci. Lett. 2010

3.1.1     Ducy et al., Cell 1997

            Komori et al., Cell 1997

            Ando et al., Curr Drug Discov Technol 2008

3.1.2     Civitelli et al., J. Clin. Invest. 1993

            Ilvesaro et al., J. Bone Miner. Res. 2000

            Yellowley J. Bone Miner. Res. 2000

            Schilling et al., J. Cell. Mol. Med. 2008

            Paznekas et al., Am. J. Hum. Genet. 2003

            Himi et al., Jpn. J. Ophthalmol 2009

            Furuta et al., Intern. Med. 2012

3.2.2     Tsai et al., Biochem. Biophys. Res. Commun. 1996

            Mehta et al., Mol. Carcinog. 1996

We hope that all the modifications made to our original manuscript address the reviewers’ issues satisfactorily and that this revised version is now acceptable for publication.

Sincerely,

Franck Verrecchia, PhD

Round 2

Reviewer 2 Report

No further comments